# 3D MOLECULAR PRETRAINING VIA LOCALIZED GEOMETRIC GENERATION

## ABSTRACT

Self-supervised learning on 3D molecular structures has gained prominence in AI-driven drug discovery due to the high cost of annotating biochemical data. However, few have studied the selection of proper modeling semantic units within 3D molecular data, which is critical for an expressive pre-trained model as verified in natural language processing and computer vision. In this study, we introduce Localized Geometric Generation (LEGO), a novel approach that treats tetrahedrons within 3D molecular structures as fundamental modeling blocks , leveraging their simplicity in three-dimension and their prevalence in molecular structural patterns such as carbon skeletons and functional groups. Inspired by masked language/image modeling, LEGO perturbs a portion of tetrahedrons and learns to reconstruct them during pretraining. The reconstruction of the noised local structures can be divided into a two-step process, namely spatial orientation prediction and internal arrangement generation. First, we predict the global orientation of the noised local structure within the whole molecule, equipping the model with positional information for these foundational components. Then, we geometrically reconstruct the internal arrangements of the noised local structures revealing their functional semantics. To address the atom-bond inconsistency problem in previous denoising methods and utilize the prior of chemical bonds, we propose to model the graph as a set of nodes and edges and explicitly generate the edges during pre-training. In this way, LEGO exploits the advantages of encoding structural geometry features as well as leveraging the expressiveness of self-supervised learning. Extensive experiments on molecular quantum and biochemical property prediction tasks demonstrate the effectiveness of our approach.

## 1 INTRODUCTION

Understanding 3D molecular structures is crucial for various tasks in drug discovery, such as molecular property prediction (Wu et al., 2018; Hu et al., 2021; Chmiela et al., 2023), binding affinity prediction (Öztürk et al., 2018; Ru et al., 2022), and docking-based generation (Ma et al., 2021; Yang et al., 2021). In recent years, self-supervised learning on 3D molecular structures has been extensively explored to learn from large collections of unlabeled compounds, which helps overcome the costly and time consuming process of annotating biochemical properties. As is demonstrated in natural language processing and computer vision, a careful selection of minimal semantic building blocks is critical for developing an expressive and robust pretrained model. By providing well-structured units, the model can effectively identify underlying patterns and extract meaningful semantics from data compositions during pretraining.

However, few existing 3D molecular pretraining methods have studied this aspect. Existing 3D molecular pretraining methods fall into two categories: representation-level and structure-level. Representation-level methods aim to enhance 2D molecular representation by leveraging information from 3D molecular structures through contrastive learning (Liu et al., 2021a; Stärk et al., 2022). Such methods use 3D molecular structures only at the encoding stage and fail to model inherent structural features through self-supervised training. Structure-level methods address this limitation by developing pre-training tasks of coordinate denoising, where independent noise is added to the coordinates of all atoms in the graph and the model is trained to reconstruct the original atomic positions (Zaidi et al., 2022; Liu et al., 2022b; Zhou et al., 2023; Jiao et al., 2023; Feng et al., 2023). However, from a chemical perspective, an atom alone can hardly serve as a functional

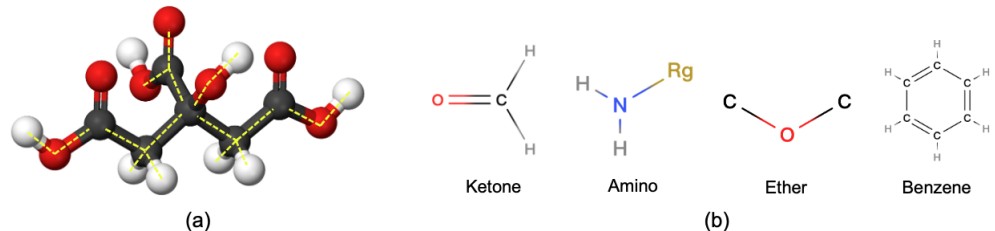

(a)          (b)

Figure 1: Local structures consisting of a central atom and its one-hop neighbors form a highly prevalent motif in molecules, which underlies (a) carbon backbones, and (b) functional groups, and etc.

unit in molecules. Therefore, atom-wise denoising provides limited improvement in the model's understanding of functional substructures.

In this paper, we focus on this open issue and propose a novel pretraining approach as an initial exploration. Our method, called Localized Geometric Generation (LEGO), treats tetrahedrons within 3D molecular structures as fundamental building blocks and tailors two pretraining tasks to learn the semantics. There are two key conceptual motivations behind this design: Geometrically, the tetrahedron is the simplest polyhedron that can be constructed in 3D Euclidean space, serving as the base case for more complex polyhedra. This structural simplicity and primitiveness aligns with the ubiquity of the tetrahedral motif in chemistry: a central atom along with its one-hop neighbors forms a highly prevalent local structure in molecules, which underlies carbon backbones, functional groups, and more (Fig 1). Therefore, tetrahedrons can be considered an excellent basic semantic unit for 3D molecular modeling from both geometry and chemistry.

Inspired by masked language/image modeling techniques (Devlin et al., 2019; Dosovitskiy et al., 2020), LEGO introduces perturbations to a portion of tetrahedrons in a 3D molecular structure and learns to reconstruct them during pretraining. In particular, we begin by segmenting a 3D molecular structure into a non-overlapping stack of one-hop local tetrahedral structures. Subsequently, we add noise or apply masks to part of the segmented local structures. The reconstruction of the perturbed local structures involves two steps: global orientation prediction and local structure generation. During the orientation prediction step, we predict the spherical coordinates of the center of mass (CoM) for each masked tetrahedron. This prediction provides positional information about local structures and their relationships within the whole molecule. While for the local generation, we introduce a geometric generation task to accurately reconstruct atom arrangements within each masked tetrahedron, which focuses on learning the pattern and semantic of the unit itself. By incorporating these steps, LEGO is able to learn both global and local features of 3D molecular geometry in a self-supervised manner.

Although the design mentioned above allows for the explicit modeling of geometric features in 3D molecular data, it is important to note that most existing 3D molecular graph models are based on nodes, where edges are represented as additional node features and not explicitly modeled. Such backbones can lead to an atom-bond inconsistency problem during the denoising-generation process generation (Peng et al., 2023). To be specific, when generating 3D structures, atom-based networks first produce atom positions and add the chemical bonds in a post-processing manner. This sequential approach may result in intermediate atom positions that are not feasible for forming bonds, leading to unrealistic topologies like extra-large ring or violate atom valency constraints. This atom-bond inconsistency presents a challenge for our pretraining approach, which focuses on reconstructing local molecular structures. In fact, bonds are critical abstract concepts in molecules as they quantify distance-dependent interaction forces between atoms and encoding key chemical semantics, and therefore play a critical role in modeling molecular local structures. To address the inconsistency, we propose modeling the molecular graph as a set of nodes and edges. During pretraining, LEGO generates the edges explicitly, allowing it to learn the significant chemical and geometric priors embedding in the bonding patterns.

The contributions of this work can be summarized as follows:

- We propose a novel self-supervised learning method for 3D molecular structures. Our approach treats tetrahedrons as the fundamental building blocks within 3D structures and introduces two pretraining tasks that enable the learning of local and global semantics in a geometric manner.

- We address the atom-bond inconsistency problem encountered in previous denoising methods by modeling the molecular graph as a set of nodes and edges. This representation leverages the prior knowledge of chemical bonds, facilitating the accurate representation of molecular structures.

- We demonstrate the effectiveness of our method through comprehensive experiments. We pretrain LEGO on a large-scale dataset and evaluate the pretrained model on biochemical and quantum property prediction tasks. The results show that our approach can well capture the molecular functional semantics and can achieve comparing results to Transformer variants with sophisticated graph-specific inductive bias.

## 2 RELATED WORKS

**3D Molecular Structure Modeling.**    3D modeling of molecular structures has been extensively explored in recent years, enabled by advancements in graph neural networks (GNN) (Wu et al., 2020; Han et al., 2022). Early work by SchNet (Schütt et al., 2017) incorporates atomic distances into continuous-filter convolutional layers to capture local atomic correlations. DimeNet (Klicpera et al., 2020) pioneers the incorporation of bond angles and directionality into vanilla GNNs, demonstrating improved performance. SphereNet (Liu et al., 2021b) and ComENet (Wang et al., 2022) introduce spherical messages to build more informative representations. To encode 3D equivariance as an inductive bias grounded in group theory, Tensor Field Networks (Thomas et al., 2018), SE(3)-Transformers (Fuchs et al., 2020) and NequIP (Batzner et al., 2022) employ tensor products, while PaiNN (Schütt et al., 2021) and EGNN (Satorras et al., 2021) adopt equivariant message passing. Beyond message passing neural networks (MPNN), the powerful transformer architecture (Vaswani et al., 2017) has also been explored for graph-structured data. Dwivedi & Bresson (2020) first introduces a fully-connected transformer for graphs and uses Laplacian eigenvectors as node positional encoding. GRPE (Park et al., 2022) and Graphormer (Ying et al., 2021) define structural positional encodings based on node topology, node-edge interaction and 3D distances. Besides positional encodings, GraphTrans (Wu et al., 2021) EGT (Hussain et al., 2022) and GraphGPS (Rampášek et al., 2022) propose hybrid architectures with stacked MPNN layers before the global attention layer. Notably, TokenGT (Kim et al., 2022) demonstrated that standard Transformers without graph-specific modifications can also achieve promising results in graph learning. Despite the success by directly incorporating 3D features into the model input, there remains a need to develop pretraining paradigms for 3D molecular structures that can learn semantic features in a self-supervised manner.

**Pretraining on 3D Molecular Structures.** Existing pre-training methods for 3D molecular structures can be categorized into two types: representation-level and structure-level. Representation-level methods use separate encoders to embed 2D graphs and 3D structures to obtain embeddings from two views, then perform contrastive learning Stärk et al. (2022) or generative self-supervised learning Liu et al. (2021a) on the two embeddings. Such methods focus on the 2D graph representation and treat 3D information as a complement to its 2D counterpart, ignoring spatial features that are more informative in determining molecular properties. Structure-level denoising tasks fill this gap by involving geometric elements in pretraining tasks. Liu et al. (2022b), Zaidi et al. (2022), Zhou et al. (2023), and Feng et al. (2023) employ denoising tasks on atomic coordinates and explore how the scale and distribution of the added noise impact the results. Zhu et al. (2022) proposes a masked modeling by predicting coordinates of masked atoms using corresponding 2D features. GEM (Fang et al., 2022) and 3D-PGT(Wang et al., 2023) use geometric features as pretraining objectives, but they implement a random masking . Different from these studies, we underscores the modeling of local semantic units in 3D molecular pretraining.

## 3 METHOD

### 3.1 MOTIVATION

Our objective is to develop a segmentation approach that effectively decomposes 3D molecular structures into suitable units for representation learning. These units need to strike a balance between two crucial factors. On one hand, the units should encapsulate the critical details related to the local molecular environment in a way that downstream models can further analyze for property predictions. On the other hand, overly complex or molecule-specific representations could limit the applicability of

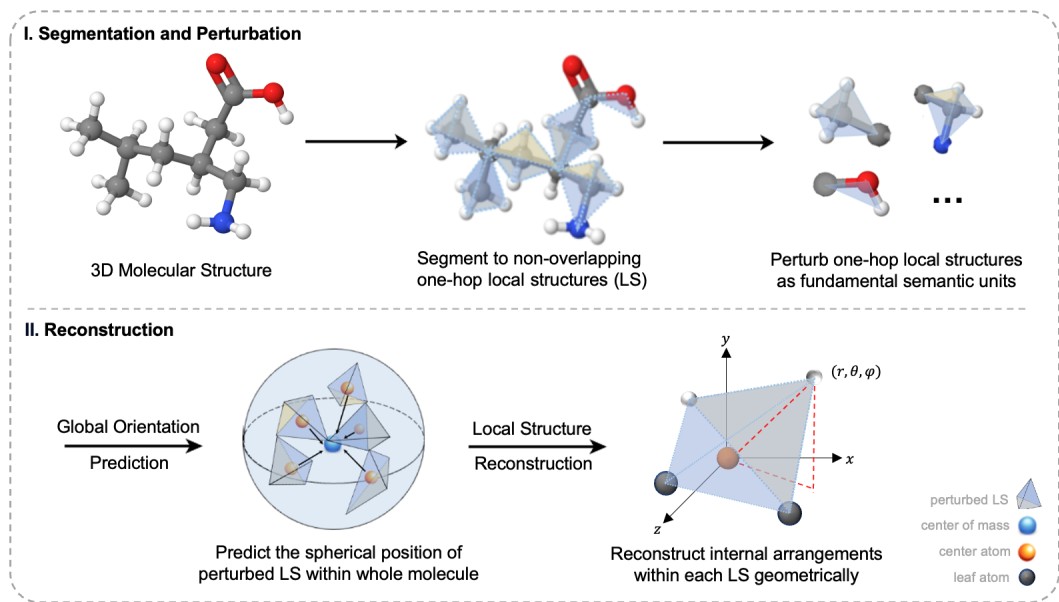

Figure 2: Overview of LEGO. **I**. Based on non-terminal atoms, we segment 3D molecular structures into building blocks of one-hop local structures (LS). We perturb a portion of the LS by adding noise to atomic positions and masking the edge features. **II**. We pre-train LEGO by geometrically reconstructing the perturbed local structures in two stages.

the approach across different chemical spaces. Therefore, we aim to identify structurally meaningful yet simple decompositions that contain rich semantics similar to how tokens and patches serve as universal elements for natural language processing and computer vision models.

Our proposed solution is to take tetrahedrons (one-hop local structures in general cases) as the fundamental building blocks. Geometrically, the tetrahedron is the simplest polyhedron that can be constructed in 3D space, serving as the base case for more complex polyhedra. This structural simplicity aligns with the widespread occurrence of the tetrahedral motif in chemical compounds, as depicted in Figure 1. In carbon skeletons and many functional groups, tetrahedral centers with a maximum valency of four allow diverse atoms to form intricate molecular structures while minimizing spatial constraints.

It is worth pointing out that the local structure of actual molecules may not always conform to a standard tetrahedral shape, and our segmentation strategy is adjusted to accommodate this variability. For center atoms with fewer than four neighbors, like the C,N,O in Fig 1(b), we simply treat the ketone, amino or the ether as a degraded tetrahedra. While for instances where center atoms form more than four bonds, such as sulfur and phosphorus, we incorporate all one-hop atoms as part of the local structure. Additionally, cyclic structures like benzene are handled by selecting non-adjacent carbons to represent the ring through a combination of its triangular fragments. By retaining this adaptive nature for atypical cases while concentrating on tetrahedra, the algorithm aims to balance simplicity and practical applicability across diverse chemical spaces.

## 3.2 TokenGT and its 3D Extension

Most existing graph neural networks typically adopt an atom-centric approach, where edge features are encoded as additional attributes and then aggregated to atoms through message passing. However, in the field of chemistry, chemical bonds play a crucial role as they abstract distance-based interatomic forces and provide essential chemical priors in local structure modeling. Neglecting the consideration of edges in molecular generation can lead to the problem of atom-bond inconsistency, resulting in the generation of undesirable molecular structures, as demonstrated by Peng et al. (2023) and Qiang et al. (2023).

In order to mitigate potential negative effects of atom-based modeling on our generative pre-training approach, In this section, we will provide a brief overview of the architecture of TokenGT and discuss a minor improvement that we propose to adapt it to 3D data.

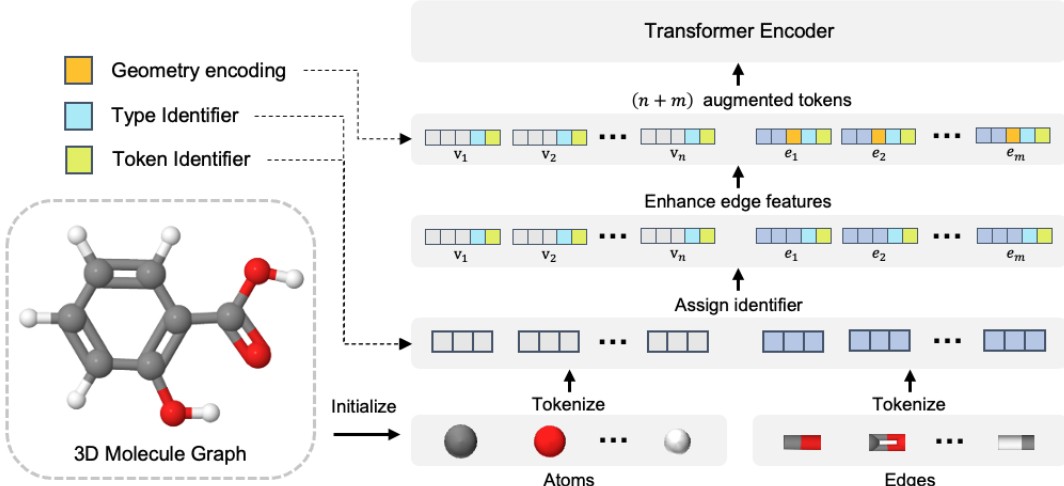

Figure 3: Architecture of the TokenGT-3D model, node and edges in graph are treated as independent tokens. Token embeddings are further augmented with graph connectivity and geometric features

**TokenGT** TokenGT, short for Tokenized Graph Transformer, has been both theoretically and empirically shown to yield promising results in graph learning. It has been demonstrated that by incorporating augmented embeddings, standard Transformers can effectively handle graph data without requiring extensive graph-specific modifications (Kim et al., 2022).

Given an input graph $\mathcal{G} = (\mathcal{V}, \mathcal{E})$, TokenGT first initializes the node set $\mathcal{V} = \{v_1, ..., v_n\}$ and the edge set $\mathcal{E} = \{e_1, ..., e_m\}$ as $\mathbf{X}^{\mathcal{V}} \in \mathbb{R}^{n \times d}$, $\mathbf{X}^{\mathcal{E}} \in \mathbb{R}^{m \times d}$. Then, each token in $\mathbf{X}$ is augmented with predefined orthonormal *token identifiers* to represent graph connectivity, and trainable *type identifiers* to encode whether a token is a node or an edge.

*Token Identifier.* Given an input graph $\mathcal{G} = (\mathcal{V}, \mathcal{E})$, $n$ node-wise orthonormal vectors $\mathbf{P} \in \mathbb{R}^{n \times d_p}$ are produced and concatenated after the token embeddings, i.e. for node $v \in \mathcal{V}$, the token $\mathbf{X}_v$ is augmented as $[\mathbf{X}_v, \mathbf{P}_v, \mathbf{P}_v]$; for edge $(u, v) \in \mathcal{E}$, the token $\mathbf{X}_{(u,v)}$ is augmented as $[\mathbf{X}_{(u,v)}, \mathbf{P}_u, \mathbf{P}_v]$.

With orthogonality, a Transformer can tell whether an edge $e = (u, v)$ is connected with a node $k$ through dot-product (attention) since $[\mathbf{P}_u, \mathbf{P}_v][\mathbf{P}_k, \mathbf{P}_k]^{\top} = 1$ if and only if $k \in (u, v)$ and $0$ otherwise. Through this design, TokenGT is able to incorporate the connectivity between nodes and edges. For more theoretical analysis of completeness and informativeness of these token identifiers, please refer to the original paper.

*Type Identifier.* Given an input graph $\mathcal{G} = (\mathcal{V}, \mathcal{E})$, TokenGT applies a trainable matrix $\mathbf{E} = [\mathbf{E}^{\mathcal{V}}; \mathbf{E}^{\mathcal{E}}] \in \mathbb{R}^{2 \times d_e}$ to augment the tokens as follows: for node $v \in \mathcal{V}$, the token $[\mathbf{X}_v, \mathbf{P}_v, \mathbf{P}_v]$ is augmented as $[\mathbf{X}_v, \mathbf{P}_v, \mathbf{P}_v, \mathbf{E}^{\mathcal{V}}]$, for edge $(u, v) \in \mathcal{E}$, the token $[\mathbf{X}_{(u,v)}, \mathbf{P}_u, \mathbf{P}_v]$ is augmented as $[\mathbf{X}_{(u,v)}, \mathbf{P}_u, \mathbf{P}_v, \mathbf{E}^{\mathcal{E}}]$.

With token identifiers and type identifiers, the initialized token embeddings $X = [\mathbf{X}^{\mathcal{V}} \in \mathbb{R}^{n \times d}, \mathbf{X}^{\mathcal{E}} \in \mathbb{R}^{m \times d}] \in \mathbb{R}^{(n+m) \times (d+2d_p+d_e)}$ are augmented to $\mathbf{X}^{in} \in \mathbb{R}^{(n+m) \times (d+2d_p+d_e)}$. Then, TokenGT passes the input to a standard Transformer encoder with vanilla multi-head self-attention layers, where a `[CLS]` token is additionally concatenated to obtain the graph embedding for downstream finetuning.

**3D Extension** To align with our geometric pretraining objectives, we propose a minor extension of the original 2D TokenGT formulation to accommodate 3D molecular graphs. Let $\mathcal{G} = (\mathcal{V}, \mathcal{E}, \mathcal{P})$ be a 3D graph, where $\mathcal{P} = \{p_1, ..., p_n\}$, $p_i \in \mathbb{R}^{n \times 3}$ is the set of atom cartesian coordinates, we augment the initial embedding $\mathbf{X}_{(u,v)}$ of edge $e_{(u,v)}$ with bond length, bond angles, and the dihedral angles realted to $e_{(u,v)}$ with a radial/spherical harmonics basis function $e_{\text{RBF}}/e_{\text{SBF}}$:

- Bond length: $\mathbf{X}^{bl}_{(u,v)} = \boldsymbol{e}^{(uv)}_{\text{RBF}} = \boldsymbol{e}_{\text{RBF}}(\|p_v - p_u\|)$

- Bond angle: $\mathbf{X}^{ba}_{(u,v)} = \sum_k \boldsymbol{a}^{(uv,uk)}_{\text{SBF}}, k \in \mathcal{N}(u) \backslash v$

- Dihedral angle: $\mathbf{X}^{da}_{(u,v)} = \sum_{k,j} \boldsymbol{a}^{(kuv,uvj)}_{\text{SBF}}, k \in \mathcal{N}(u) \backslash v, \ j \in \mathcal{N}(v) \backslash u$

- Augmented edge embedding: $\mathbf{X}^{\text{3D}}_{(u,v)} = \mathbf{X}_{(u,v)} + \mathbf{X}^{bl}_{(u,v)} + \mathbf{X}^{ba}_{(u,v)} + \mathbf{X}^{da}_{(u,v)}$

---

**Algorithm 1** Local Structure Reconstruction in LEGO

---

**Require:**
$\mathcal{G}$: Input graph $\mathcal{G} = (\mathcal{V}, \mathcal{E}, \mathcal{P})$ with $n$ nodes and $m$ edges.
$\mathbf{M}_{\text{center}} = \delta^n, \mathbf{M}_{\text{edge}} = \delta^m, \mathbf{M}_{\text{leaf}} = \delta^n, \delta \in \{0, 1\}$: Mask indicators for center atoms, edges, leaf atoms.
$\mathbf{Emb}^{(n+m) \times \dim}$: Embedding for tokens in $\mathcal{G}$ after a standard Transformer encoder.
$\text{LEGOHead}_i, i \in \{1, 2, 3, 4\}$: Network module for reconstructing perturbed local structures. The four values of $i$ correspond to global orientation of center atoms, edge length of edges, azimuthal angles of leaf nodes, and polar angles of leaf nodes, respectively.
**Labels**: Ground truth labels of the geometric elements: $z, l, \theta, \phi$.
$T$: Training Steps

1: **while** $T \neq 0$ **do**
2:     Pad $\mathbf{M}_{\text{center}}, \mathbf{M}_{\text{edge}}, \mathbf{M}_{\text{leaf}}$ to size $[n + m, 1]$
3:     $z^{pred} = \text{LEGOHead}_1(\mathbf{Emb}[\mathbf{M}_{\text{center}}])$
4:     $l^{pred} = \text{LEGOHead}_2(\mathbf{Emb}[\mathbf{M}_{\text{edge}}])$
5:     $\theta^{pred} = \text{LEGOHead}_3(\mathbf{Emb}[\mathbf{M}_{\text{leaf}}])$
6:     $\psi^{pred} = \text{LEGOHead}_4(\mathbf{Emb}[\mathbf{M}_{\text{leaf}}])$
7:     Loss $= w_{\text{distance}} \cdot \text{MSELoss}(\mathbf{Labels}, z^{pred}, l^{pred}) + w_{\text{angle}} \cdot \text{VonMisesLoss}(\mathbf{Labels}, \theta^{pred}, \psi^{pred})$
8:     Optimise(Loss)
9:     $T = T - 1$
10: **end while**

---

## 3.3 Pretrain via Localized Geometric Generation

At a high level, our method first segments the 3D molecular structure into non-overlapping, one-hop local structures. We then perturb a proportion of these units through a corruption strategy that masks token attributes and adds noise to node coordinates simultaneously. Subsequently, we reconstruct the perturbed local structures in a generative way by predicting their global orientation and local geometric arrangements. Figure2 visualizes the workflow of our method.

**Local Structure Segmentation** The core idea of local structure segmentation is to ensure none of the segmented results should be overlapped, that is to say, a leaf node in one local structure cannot be the center node in another local structure, but the overlapping of two leaf nodes is allowed. To elaborate, we first traverse the graph nodes in a BFS order $\pi$, collect the non-terminal nodes as $\mathcal{V}_{\text{non-terminal}}$, and initialize a boolean tensor $f_{\text{segmented}} = \mathbf{0}^{\top}$. Then, we sample a node $u$ from $\mathcal{V}_{\text{non-terminal}}$ to form a local structure, where we add $u$ to $\mathcal{V}_{\text{seg-center}}$ and set the flags of its one-hop neighbors to true $f_{\text{segmented}}[v] = \text{True}, v \in \mathcal{N}(u)$. We then repeat the above operation until all the atoms in $\mathcal{V}_{\text{non-terminal}}$ have been segmented.

Though our segmentation algorithm possesses randomness and may leave out terminal atoms at times, we see it as a way to increase the generalizability and robustness. By sampling different central nodes during segmentation, the model is encouraged to learn more holistic representations rather than relying on a fixed decomposition across multiple pretraining iterations. Regarding terminal atoms that are initially excluded from segmented units, they are likely to be eventually incorporated through successive iterations that segment their tetrahedron-like neighborhoods.

**Local Structure Perturbation** Given the segmented result of a molecular graph $\mathcal{V}_{\text{seg-center}}$, we randomly perturb some local structures with ratio $m_{\text{LS}}$ and get the set of masked centers $\mathcal{V}_{\text{mask-center}}$ and an indicator tensor $\mathbf{M}_{\text{center}} = \{0, 1\}^n$. Since we mask all the nodes and edges in the selected local structures, the mask ratio over all tokens (atoms and edges) $m_{\text{token}}$ will be different from $m_{\text{LS}}$, which statistical relationship between the two mask ratio is in displayed in AppendixA. Based on the masked centers, we can denote the rest of the perturbed local structures as $\mathcal{E}_{\text{mask-edge}} = \{(u, v) | u \text{ or } v \in \mathcal{V}_{\text{mask-center}}\}$, and $\mathcal{V}_{\text{mask-leaf}} = \{v | (u, v) \in \mathcal{E}_{\text{mask-edge}} \text{ for } u \in \mathcal{V}_{\text{mask-center}}\}$, along with $\mathbf{M}_{\text{edge}} \in \{0, 1\}^m$ and $\mathbf{M}_{\text{leaf}} \in \{0, 1\}^n$. Then, we conduct perturbation by adding coordinate noise to atoms in $\mathcal{V}_{\text{mask-center}}$ and $\mathcal{V}_{\text{mask-leaf}}$, as well as masking the edge attributes in $\mathcal{E}_{\text{mask-edge}}$.

**Local Structure Reconstruction** To successfully reconstruct the perturbed local structures, we must consider two critical aspects: the global orientation of the local structure within the entire molecule and the internal arrangements between nodes and edges within a local structure.

Table 1: Results for biochemistry property prediction tasks. We compare our models with existing 2D or 3D molecular pretraining models. The best and second best results are **bold** and underlined.

| model | Classification (ROC-AUC ↑) | | | | | Regression (MAE ↓) | | |
|---|---|---|---|---|---|---|---|---|
| | BACE | BBBP | Clintox | SIDER | Tox21 | Freesolv | Esol | Lipo |
| AttrMask (Hu et al., 2019) | 84.5 | 68.7 | 72.6 | 62.7 | 78.1 | 2.764 | 1.100 | 0.739 |
| GROVER (Rong et al., 2020) | 81.0 | 69.5 | 76.2 | 65.4 | 68.2 | 2.272 | 0.895 | 0.823 |
| MolCLR (You et al., 2020) | 82.4 | 72.2 | 91.2 | 58.9 | 75.0 | 2.594 | 1.271 | 0.691 |
| 3DInfomax (Stärk et al., 2022) | 79.4 | 69.1 | 9.4 | 53.3 | 74.4 | 2.337 | 0.894 | 0.695 |
| GraphMVP (Liu et al., 2021a) | 81.2 | 72.4 | 79.1 | 63.9 | 75.9 | - | 1.029 | 0.681 |
| GEM (Fang et al., 2021) | **85.6** | 72.2 | 90.1 | 67.2 | 80.6 | 1.877 | 0.798 | 0.660 |
| Uni-Mol (Zhou et al., 2023) | **85.6** | 72.4 | 91.9 | 65.9 | 79.6 | **1.620** | 0.788 | **0.603** |
| 3D PGT (Wang et al., 2023) | 80.9 | 72.1 | 79.4 | 60.6 | 73.8 | - | 1.061 | 0.687 |
| LEGO | 81.9 | **74.2** | **94.3** | **72.3** | **83.9** | 1.844 | **0.704** | 0.804 |

Regarding spatial orientation, we predict the spherical coordinates of central atoms within masked local structures. These coordinates indicate where to position each unit within the overall molecule and its orientation relative to other units. For internal geometry, the previously predicted central atom serves as the origin of a spherical coordinate system (SCS). We then predict the radial distance ($r$, edge length), azimuthal angle ($\theta$), and polar angle ($\psi$) of each masked peripheral atom within this SCS. Edge lengths are directly predicted as they closely relate to bond type. Meanwhile, angular values guide subsequent reconstruction of three-dimensional coordinates for the peripheral atoms. The procedure of the local structure reconstruction of our method is summarized in Algorithm 1.

We use Mean Squared Error as the loss function for edge length and radius, and adopt the von Mises-Fisher Loss to train angle-related terms.

## 4 EXPERIMENTS

### 4.1 DATASETS AND EXPERIMENTAL SETUP

**Pre-training.** We pretrain LEGO on OGB-PCQM4Mv2 dataset Hu et al. (2021), which contains 3D molecular structures simulated by density functional theory (DFT). The dataset has 3.38 million molecules, each with one dominant equilibrium conformation. While considering multiple conformations can describe 3D molecular structures more comprehensively and improve representability (Liu et al., 2021a; Stärk et al., 2022), we believe that learning molecular semantics from the dominant conformation is sufficient to validate our method. Handling multiple conformations is left for future work.

We follow the Transformer encoder configuration from the original TokenGT base model: 12 layers, 768 embedding dimension, 32 attention heads and use Graph Laplacian as the node identifier. We mask $m_{\text{LS}}$=10% of the local structures and set the noise scale on coordinate noise to 0.3. The weights for distance loss $w_{\text{distance}}$ and angle loss $w_{\text{angle}}$ are both set to 1. We use AdamW optimizer with $(\beta_1, \beta_2) = (0.99, 0.999)$ and a weight decay of 0.1. We apply the polynomial learning rate scheduler, with a peak learning rate of 2e-4 and 150k warm-up steps over 1M iteration with a batch size 256. The model is pretrained on 8 NVIDIA A100s for 300 epochs.

**Fine-tuning.** We use the `[CLS]` token as the graph representation for downstream finetuning and pass it through a two-layer MLP projection head for task predictions. We evaluate the pretrained model on biochemical and quantum molecular properties. Biochemical properties test how well the model captures semantics from the segmented units within a molecule, while quantum properties test the model's ability to represent 3D structures in terms of interatomic interactions.

For biochemical properties, we choose the widely-used benchmark MoleculeNet Wu et al. (2018), where the related tasks can be categorized into physical chemistry, biophysics, and physiology. The original MoleculeNet dataset contains only 2D data and existing 3D pretraining baselines take 2D graph as input as well. We follow this setting to demonstrate the transferability of our pretrained model.

Table 2: Results on PCQM4Mv2 validation set in OGB Large-Scale Challenge Hu et al. (2021). The results are evaluated by Mean Absolute Error (MAE). The best and second best results are **bold**.

| model | #param. | Valid MAE ($\downarrow$) |
|---|---|---|
| GraphGPS$_{SMALL}$ (Rampášek et al., 2022) | 6.2M | 0.0938 |
| GRPE$_{BASE}$ (Park et al., 2022) | 46.2M | 0.0890 |
| EGT (Hussain et al., 2022) | 89.3M | 0.0869 |
| GRPE$_{LARGE}$ (Park et al., 2022) | 46.2M | 0.0867 |
| Graphormer (Ying et al., 2021) | 47.1M | 0.0864 |
| GraphGPS$_{BASE}$ (Rampášek et al., 2022) | 19.4M | 0.0858 |
| GraphGPS$_{DEEP}$ (Rampášek et al., 2022) | 13.8M | 0.0852 |
| GEM-2 (Liu et al., 2022a) | 32.1M | 0.0793 |
| Transformer-M (Luo et al., 2022) | 47.1M | 0.0787 |
| GPS++$_{BASE}$ (Masters et al., 2022) | 44.3M | 0.0778 |
| 3D GPT Wang et al. (2023) | 42.6M | **0.0762** |
| TokenGT (Kim et al., 2022) | 48.5M | 0.0910 |
| LEGO (ours) | 52.7M | 0.0817 |

Following previous works Zhu et al. (2022); Fang et al. (2022), the datasets are splitted according to their molecular scaffolds by 8:1:1. We use bayesian search to find the best hyper-parameter combination with a maximum trials of 64.

For quantum properties, we choose the OGBLSC-PCQM4Mv2 (Hu et al., 2021) as the benchmark. Given 3D molecular structures, the task requires the model to predict the HOMO-LUMO gap of the molecules, an important quantum property that has been shown to closely correlate with macro molecular properties. Since the test set is not open-sourced, we report the validation MAE as the result as most methods do.

**Baselines.** For MoleculeNet, we mainly compare LEGO with existing state-of-the-art 3D-based pretrained models in Stärk et al. (2022); Liu et al. (2021a); Fang et al. (2022); Zhu et al. (2022). We also select three typical pretraining models on 2D graphs in order to illustrate the effectiveness of leveraging 3D geometry information: AttrMask (Hu et al., 2019), GROVER (Rong et al., 2020), and GraphCLR (You et al., 2020).

In terms of quantum property prediction, our baselines cover the currently SOTA methods, including GraphGPS (Rampášek et al., 2022), GRPE (Park et al., 2022), EGT (Hussain et al., 2022), Graphormer (Ying et al., 2021), Transfomer-M (Luo et al., 2022), GPS++ Masters et al. (2022) and 3D-GPT (Wang et al., 2023).

## 4.2 MAIN EXPERIMENTAL RESULTS

In this section, we evaluate our pretrained model on the two property prediction tasks and analyse what improvement the model can obtain via our structured pretraining.

For biochemical properties, we achieve state-of-the-art results on 5 out of 8 tasks and comparable performance on 2 additional tasks (Table 1). Specifically, LEGO demonstrates significantly improved performance on predicting physiological properties like toxicity, indicating that our method can effectively capture functional semantics in molecular structures. LEGO also achieves strong results on tasks such as Freesolv and Esol, which are related to the properties of molecules in a water environment. However, it underperforms on Lipo, which is related to a lipid environment. This difference in transfer learning may be due to the significant difference between the conformations molecules exhibit in a lipid environment and the equilibrium conformations used in our pretraining. Again, these results validate our motivation that exploiting functional semantics through proper segmentation of molecular structures is vital.

Table 2 exhibits the validation results on PCQM4M-v2 for quantum property prediction. As shown in the table, although LEGO boosts the performance with 10.2% over the non-pretrained TokenGT, it lags behind the state-of-the-art result. However, we would like to argue this is because all the other baselines are introducing complicated graph-specific encodings into the model, while we utilize a

Table 3: Ablation results on PCQM4M-v2 for different $m_{LS}$ and noise scales.

| $m_{LS}$ | noise scale | equivalent $m_{atom}$ | Valid MAE |
|---------|-------------|-----------------------|-----------|
| 0.1 | 0.3 | 0.36 | **0.0817** |
| 0.1 | 1.0 | 0.36 | 0.0862 |
| 0.15 | 0.3 | 0.57 | 0.0877 |
| 0.2 | 0.3 | 0.77 | 0.0885 |

pure transformer backbone. The primary contribution of this work is to give a glimpse at how proper selection of semantic units impacts 3D molecular pretraining, and we believe a further introduction of graph inductive bias will further improve our result.

### 4.3 ABLATION STUDIES

In this section, we ablate key design elements of the proposed LEGO pretraining paradigm.

**Mask Ratio and Noise Scale**  In Zaidi et al. (2022) and Feng et al. (2023), the authors point out that in molecular denoising pretraining, excessive noise often leads to training divergence and detrimental impacts. Will this conclusion still hold on our structured pretraining? The ablation results in Table 3 give a positive answer. From the table, we observe decreased performance on PCQM4M-v2 as the mask ratio and noise scale parameters for local structure (LS) perturbation are increased. We attribute this trend to greater difficulty in reconstructing the original data when more extensive corruption is introduced across larger molecular fractions during pre-training. Specifically, higher mask ratios lead to a greater number of perturbed local structures, while larger noise scales further distort the original topology of the units. With excessive corruption, preserving original structural semantics for reconstruction becomes more challenging, limiting gains from the pre-training phase for downstream transfer.

**Random vs Structured**  To ablate the effect of our structured design in pretraining, we adopt a random masking on atoms with $m_{atom} = 0.36$, which corresponds to its structured counterpart $m_{LS} = 0.1$. Table 4 demonstrate that naive atomic-level noise leads to inferior performance compared to LEGO's incorporation of structural semantics during perturbation and reconstruction, quantifying the consequent gains of a chemistry-aware, structure-based procedure for molecular representation enhancement through self-supervised objectives.

Table 4: Comparison for random and structured pretraining on PCQM4M-v2.

| model | Valid MAE |
|-------|-----------|
| LEGO | **0.0817** |
| randomly perturbed | 0.0883 |

## 5 CONCLUSION

In this paper, we propose a novel approach for self-supervised learning on 3D molecular structures. By treating tetrahedrons within 3D molecular structures as fundamental building blocks, we implement structured denoising to capture both local and global features. We also address the atom-bond inconsistency problem by explicitly modeling edges in molecular graph. Through pretraining, our approach achieves competitive results on both biochemical and quantum molecule property prediction tasks. In the future, we aim to investigate integrating additional graph inductive biases into the model while retaining explicit edge representations. Furthermore, we plan to validate the proposed segmentation strategy across a broader range of molecular structures and explore alternate perturbation techniques.

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

# A    STATISTICS BETWEEN $m_{\text{LS}}$ AND $m_{\text{TOKEN}}$

In our perturbing strategy, we select local structures to perturb with the mask ratio $m_{\text{LS}}$ on local structures. When one local structure is selected, all the atoms and edges within will all be perturbed, making $m_{\text{LS}}$ a different metric compared to the mask ratio on language tokens or image patches. For a more intuitive display, we present the corresponding statistics between $m_{\text{LS}}$ and $m_{\text{token}}$ in TableA.

As we can see from the table, a $m_{\text{LS}}$ of 0.25 will mask nearly 90% atoms in a molecule, and increasing $m_{\text{LS}}$ from 0.3 to 0.5 make no changes. In our implementation, we choose 0.10 as $m_{\text{LS}}$, which corresponds to 35% masking ratio on atoms and 21% masking ratio on tokens.

Table 5: Statistics between $m_{\text{LS}}$ and $m_{\text{token}}$

| $m_{\text{LS}}$ | 0.05 | 0.10 | 0.15 | 0.20 | 0.25 | 0.30 | 0.50 |
|---|---|---|---|---|---|---|---|
| LS Masked per Molecule | $1.07_{(0.25)}$ | $2.50_{(0.74)}$ | $3.94_{(1.06)}$ | $5.32_{(1.32)}$ | $6.00_{(1.42)}$ | $6.15_{(1.44)}$ | $6.15_{(1.44)}$ |
| Neighbor Atoms Masked per LS | $3.28_{(0.69)}$ | $3.28_{(0.49)}$ | $3.28_{(0.41)}$ | $3.28_{(0.37)}$ | $3.28_{(0.35)}$ | $3.28_{(0.35)}$ | $3.28_{(0.35)}$ |
| Atoms Masked per Molecule | $4.60_{(1.47)}$ | $10.82_{(3.75)}$ | $17.04_{(5.44)}$ | $22.95_{(6.64)}$ | $25.78_{(6.79)}$ | $26.35_{(6.71)}$ | $26.35_{(6.71)}$ |
| Edges Masked per Molecule | $3.53_{(1.25)}$ | $8.32_{(3.05)}$ | $13.10_{(4.42)}$ | $17.63_{(5.38)}$ | $19.77_{(5.45)}$ | $20.20_{(5.36)}$ | $20.20_{(5.36)}$ |
| Edges Mask Ratio per Molecule | $0.06_{(0.02)}$ | $0.14_{(0.03)}$ | $0.22_{(0.04)}$ | $0.29_{(0.04)}$ | $0.33_{(0.04)}$ | $0.34_{(0.05)}$ | $0.34_{(0.05)}$ |
| Equivalent Mask Ratio Distribution | $0.1612_{(0.05)}$ | $0.3598_{(0.07)}$ | $0.5682_{(0.08)}$ | $0.7702_{(0.09)}$ | $0.8742_{(0.11)}$ | $0.8983_{(0.13)}$ | $0.8983_{(0.13)}$ |
| Equivalent Token Mask Ratio Distribution | $0.0942_{(0.03)}$ | $0.2101_{(0.04)}$ | $0.3317_{(0.05)}$ | $0.4495_{(0.05)}$ | $0.5096_{(0.07)}$ | $0.5234_{(0.08)}$ | $0.5234_{(0.08)}$ |

