# OpenReview forum: "3D Molecular Pretraining via Localized Geometric Generation"
_ICLR.cc/2024/Conference — Submitted to ICLR 2024_

### Official Review · Reviewer_wqqh · 2023-10-29

**Soundness:** 3 good
**Presentation:** 3 good
**Contribution:** 2 fair
**Rating:** 3
**Confidence:** 5

**Summary:**

This paper introduces a 3D molecular self-supervised learning approach that leverages the geometric information of molecular local structures, in the way of orientation prediction and arrangement generation. The atom-bond inconsistency issue has been identified and tackled through a joint modeling of the graph as a set of nodes and edges. The method has been benchmarked on MoleculeNet and OGBLSC-PCQM4Mv2 datasets to verify the efficacy of proposed designs.

**Strengths:**

1. The paper is well-motivated through the concept of molecular local structure and the introduced approach yields good novelty.

2. The presentation is mostly clear and the method is easy to follow.

**Weaknesses:**

1. The experimental results seem to be insufficient to support the empirical superiority of the proposed approach. In particular, the method could be further improved either through a more careful design of the backbone or enhancements of the training objective to make the results stronger.

2. If the results are difficult to improve, the authors may also be suggested to try other benchmarks or setups, e.g., QM9, MD17, where a better utilization of the 3D structural information would bring more benefits. In the current shape the quality of the experimental evaluations may not meet the bar of ICLR.

**Questions:**

Q1. I am curious whether the proposed SSL objective can be combined with other backbones or even other pretraining objectives. If so, it would be interesting to see how the method can benefit different backbones which may be an evidence of the extensibility of the approach.

Q2. How does the method perform on datasets like QM9?

Q3. For the ablation study, it would also be interesting to see how the proposed 3D TokenGT helps to boost the performance since one of the claim in the paper is that the atom-bond inconsistency problem is tackled by modeling graph as a set of nodes and edges.

---

### Official Review · Reviewer_SwR4 · 2023-10-30

**Soundness:** 2 fair
**Presentation:** 2 fair
**Contribution:** 2 fair
**Rating:** 3
**Confidence:** 4

**Summary:**

This paper describes a pretraining process on the 3D structure of small molecules. The method splits a molecule into tetrahedrons of a few atoms each and learns to reconstruct them in 3D using knowledge of the graph structure of the input molecule. The authors apply the model on downstream property prediction tasks and compare to public benchmarks.

**Strengths:**

The main method in the paper is interesting conceptually and somewhat original as it combines a graph token representation with a decomposition of a small molecule into smaller 3D structural units. The downstream application on MoleculeNet is significant, and the application on the open graph benchmark is useful for context, although the particular table 2 is misleading. The introduction is reasonably clear, however, other parts of the paper have problems with the language or formalism.

**Weaknesses:**

The paper can use a lot of rewriting in the methods and experimental results sections.


Table 1 does not correctly highlight the best and second best results (see column for BACE when 2 of the 2D models perform better than LEGO).  Table 2 is missing the vast majority of best performing models from the opengraph benchmark large scale challenge.  Interestingly, the comparison in table 2 misses the top two entries which are models included in Table 1; if I count correctly the LEGO model would rank 9th in the validation metric with a substantial gap compared to unimol published in last year's ICLR.  It would be useful if the authors submitted their model to the benchmark to see the performance on the test set.

The text has a lot of rushed / unclear sentences.  The first sentence on page 8 ("All this baselines involves...") does not make sense as written.  Small errors and lack of clarity starts earlier in page 5 (undefined d_p), page 6 ("way to increase the and generalizability"), page 7 ("... is enough to valid our method", "...graph representation an pass it...", "...an important properties...", "...proved to be close related..."), probably more.

**Questions:**

Although the current method is not directly inspired by this work, I believe that the RL reconstruction of molecular geometries from 3D fragments in Flam-Shepherd et al (https://arxiv.org/abs/2202.00658) relates closely to the core inspiration of this method and might warrant discussion in the intro. (In contrast, some of the discussion of the atom-bond inconsistency problem is potentially possible to skip as it doesn't add meaningful insights.) Did the authors think of extending their tetrahedral segmentation approach to use a similar fragment-based approach instead?

Can the authors comment on the disparity of the approach of their validation scores on moleculeNet vs those on the large-scale challenge?

The ablation study is not helpful: the perturbation of the model is too limited, and table 3 suggests strongly that the parameters are actually not optimal.  The random perturbation pretraining of table 4 is not described in a clear enough fashion.

---

> ### Author Response · Authors · 2023-11-23
> **Response to Reviewer SwR4**
>
> Thank you for the thorough review and positive feedback - your suggestions will lead to a stronger analysis. To address your concern, we provide the following point-to-point responses.
>
> >   **Q1**:...... I believe that the RL reconstruction of molecular geometries from 3D fragments in Flam-Shepherd et al (https://arxiv.org/abs/2202.00658) relates closely to the core inspiration of this method and might warrant discussion in the intro....
>
> **Response**: We appreciate you highlighting the promising fragment-based reconstruction approach by Flam-Shepherd et al. As you suggested, we will cite this relevant work in the introduction and discuss extensions of our tetrahedron segmentation to incorporate larger functional fragments as building blocks. This could enable more complex biologically meaningful reconstruction objectives in our future work.
>
> >   **Q2**: Can the authors comment on the disparity of the approach of their validation scores on moleculeNet vs those on the large-scale challenge?
>
> **Response:** For PCQM4M-v2, we would like to argue that existing top approaches incorporate sophisticated graph encoding priors, while we apply a pure transformer. Nevertheless, the benefits of our pretraining approach can be demonstrated by the performance increase over the non-pretrained TokenGT model.  The primary contribution of this work is to give a glimpse at how proper selection of semantic units impacts 3D molecular pretraining, and we believe a further introduction of graph inductive bias will further improve our result in table2.
>
> >   **Q3:** The ablation study is not helpful: the perturbation of the model is too limited, and table 3 suggests strongly that the parameters are actually not optimal. The random perturbation pretraining of table 4 is not described in a clear enough fashion.
>
> **Response:** Thank you for the feedback requesting more extensive ablation experiments and analysis, and we have revised the abaltion study section in our paper. In table 3, we show that excessive noise in 3D structure denoising can lead to training divergence and detrimental impacts and validate the setting of our perturbation strategy. In table 4, we validate the benefits of structural pretraining by comparing against a naive random atom-level denoising baseline with equivalent 36% masking.

---

### Official Review · Reviewer_2SQj · 2023-11-01

**Soundness:** 2 fair
**Presentation:** 3 good
**Contribution:** 2 fair
**Rating:** 3
**Confidence:** 4

**Summary:**

This paper proposes a self-supervised pre-training strategy for 3D molecular structures, based on partitioning molecular structure into tetrahedra that can then be masked and reconstructed.

**Strengths:**

The empirical performance is OK for some of the fine tuning tasks, although still not consistently solid.

**Weaknesses:**

The geometrical justification of tetrahedra as simplest polyhedron might make sense, but in chemistry it makes a lot less sense. The authors literally show benzene in Figure 1b, which has 120-degree bonding pattern (so called planar-trigonal in chemistry) that is NOT a tetrahedron at all, and has very different local symmetries.
The ablation studies are unconvicing. Why not evaluate the role of the actual innovations introduced ? Tetrahedra vs. point-wise generation. Evaluate the role of edge information ?
What happens for atoms with more than 4 bonds (sulfur, phosphorous, etc) ?

**Questions:**

"We attributethistothedifferent3Dstructuresmoleculesexhibitinliposomecompounds." What does this mean ? What are these structures different ?
All the biochemistry prediction tasks are actually properties of the graph, not the 3D structure, What 3D structure is being used ?

---

> ### Author Response · Authors · 2023-11-23
> **Response to Reviewer 2SQj**
>
> We really appreciate the helpful comments and critiques that allow us to improve LEGO. Regarding your concern, here is our response.
>
> >   **Q1:** The geometrical justification of tetrahedra as simplest polyhedron might make sense, but in chemistry it makes a lot less sense. The authors literally **show benzene in Figure 1b, which has 120-degree bonding pattern (so called planar-trigonal in chemistry)** that is NOT a tetrahedron at all, and has very different local symmetries
> >
> >   ......
> >
> >   What happens for atoms with more than 4 bonds (sulfur, phosphorous, etc) ?
>
> **Response**: Thank you for highlighting this important point. We have expanded the motivation section (page 4) to better address cases where local molecular structures deviate from standard tetrahedral geometry. While tetrahedra serve as an informative reference decomposition target due to their simplicity, our segmentation strategy adapts accordingly when this arrangement does not naturally arise.
>
> For instance, center atoms with fewer than four one-hop neighbors, as commonly observed in functional groups (e.g. ketenes, amines, ethers in Fig. 1b), are handled by treating the local unit as a degraded tetrahedra. Meanwhile, atoms forming more than four bonds, like sulfur and phosphorous, we incorporate all its one-hop neighbors into the unit. For cyclic structures such as benzene, non-adjacent ring atoms are selected, effectively partitioning the system into triangular fragments. In summary, while tetrahedral local structures provide a useful illustration, our method maintains flexibility to accommodate chemical environments where alternative local symmetries dominate.
>
> >   **Q2**: "We attribute this to the different 3D structures molecules exhibit in liposome compounds." What does this mean ? What are these structures different ?
>
> **Response**:  Thank you for raising this question. To clarify, when referring to the "different 3D structures exhibited by molecules in liposome compounds", we intended to highlight potential differences between conformations in a lipid environment versus the conformations in a water environment, or the equilibrated 3D conformations used during pre-training. As you pointed out, our original phrasing was unclear on this distinction.
>
> We hypothesize that the weaker performance on the Lipo task may arise because the stable conformations adopted by molecules in a lipid-like environment can differ substantially from common isolated conformations. As our pre-training exclusively saw the latter, transferring representations to predict lipid-specific properties poses an additional challenge.
>
> >   **Q3:** All the biochemistry prediction tasks are actually properties of the graph, not the 3D structure, What 3D structure is being used ?
>
> Response: We would like to point out that we adopt only 2D graph inputs for the biochemistry prediction tasks to **conform with** established dataset conventions and existing baseline approaches on MoleculeNet, including GraphMVP, 3DInfomax, and other 3D pre-trained models - all of which use 2D inputs. As such, while not directly evaluating 3D structure modeling, these tasks probe valuable model capabilities: 1.Encoding functional semantics related to downstream biological effects into 2D topology representations. 2.Assessing transferability of features learned from 3D pre-training when adapted to 2D inputs.
>
> Conversely, we intentionally select 3D-sensitive quantum property prediction as an additional testbed for explicit 3D molecular structure modeling, using true 3D conformations as inputs. This directly examines the model's proficiency at encoding geometric details within continuous predictions - complementing the more functional evaluation on biochemical properties.
>
> ####

---

### Official Review · Reviewer_VdE4 · 2023-11-02

**Soundness:** 2 fair
**Presentation:** 3 good
**Contribution:** 2 fair
**Rating:** 3
**Confidence:** 4

**Summary:**

The paper presents a novel pre-training approach for 3D molecular datasets, building upon TokenGT. By augmenting TokenGT with 3D attributes, the proposed method harnesses 3D molecular data to enhance its generalization capabilities across various tasks.

**Strengths:**

1. Segmenting 3D structures into distinct local components is an insightful approach.

2. The reconstruction method for these local structures is intriguing and presents a unique strategy.

3. There is a marked novelty in the concept of pre-training 3D molecules through local structures, despite the base model being a straightforward extension of TokenGT enhanced with 3D coordinates.

**Weaknesses:**

1. The base model's simple strategy of appending 3D features to token embeddings may compromise the essential equivariance of 3D molecules.

2. The description of the training objective section lacks clarity. I have outlined specific inquiries in the subsequent Questions.

3. Given that the model is training on the reconstruction of molecular conformations, it would be beneficial to disclose the reconstruction accuracy on the pretraining dataset to demonstrate the model's learning efficacy.

4. The experimental comparisons should encompass additional 3D molecular datasets such as QM9 and GEOM-drug, considering the model's pretraining on 3D structures. Nonetheless, the paper confines its reporting to the OGBLSC-PCQM4Mv2 dataset and the observed performance significantly lags behind SOTA methods. The explanation provided for this underperformance does not sufficiently account for these results.

5. The layout of the paper requires revision. The Algorithm should be positioned before page 10 or on a separate page designated as an appendix, rather than following the references on page 12.

**Questions:**

1. What exactly constitutes the input for TokenGT-3D? Is it the original molecules, or do you utilize each local structure segmentation after masking and perturbation? Or is it the masked and perturbed local structure segmentations of a single molecule, or something else entirely?

2. Given the proposed method centers on pre-trained representation learning, what form do the learned representation embeddings take for downstream tasks?

3. Could you elaborate on how the local structures are reconstructed? What serves as the input for this process: a single embedding from the TokenGT-3D output, or a collection of embeddings from local structure segmentations within a single molecule?

---

> ### Author Response · Authors · 2023-11-23
> **Response to Reviewer VdE4**
>
> We sincerely thank you for the valuable remarks that help us better examine our work. To address your concern, we present the following responses.
>
> >   **Q1**: **What exactly constitutes the input for TokenGT-3D?** Is it the original molecules, or do you utilize each local structure segmentation after masking and perturbation? Or is it the masked and perturbed local structure segmentations of a single molecule, or something else entirely?
>
> **Response:** During pretraining, TokenGT-3D still take node and edge tokens, i.e. the original molecules as the input, as is shown in Figure 3 on page 4. The attribute mask and coordinate noise at perturbation stage are implemented token-wise during perturbation. Afterwards, the model takes both clean and perturbed tokens of the original molecule as the input; the segmentation on molecular conformations does not function as a 3D tokenizer.
>
> >   **Q2**: Given the proposed method centers on pre-trained representation learning, **what form do the learned representation embeddings take for downstream tasks**?
>
> **Response**:  A $[\texttt{CLS}]$ token is concatnated after the augmented node and edge embedding before the forward process of the Transformer encoder and serve as the graph representation for downstream tasks. This BERT-style method follows the implementation in the original TokenGT.
>
> >   **Q3**: Could you elaborate on **how the local structures are reconstructed**? What serves as the input for this process: a single embedding from the TokenGT-3D output, or a collection of embeddings from local structure segmentations within a single molecule?
>
> **Response**: Thanks for your feedback. To better illustrate the local structure reconstruction procedure, we have revised the paper to reposition Algorithm 1 within the methodology section. As is previously stated in Q1, the model takes both clean and perturbed tokens as input, producing an embedding $\mathbf{Emb}$ in shape $[n+m,dim]$. During pretraining, we utilize three separate mask indicators -- $M_\text{center}, M_\text{edge},M_\text{leaf}$ -- for center atoms, edges and leaf atoms within the whole molecule. For reconstruction, the central atom embeddings, $\mathbf{Emb}[M_\text{center}]$, are fed into a specialized head to predict global orientation. Likewise, edge embeddings $\mathbf{Emb}[M_\text{edge}]$ and leaf node embeddings $\mathbf{Emb}[M_\text{leaf}]$are input into dedicated modules for reconstructing edge lengths and angles, respectively.

---

### Meta-Review · Area_Chair_qcpZ · 2023-12-05

**Metareview:**

The paper introduces a novel self-supervised pre-training method for 3D molecular datasets. This method leverages the geometric information of local molecular structures by dividing the molecule into tetrahedra of a few atoms and using a graph-based representation. The authors then apply the model to downstream property prediction tasks and compare it to publicly available benchmarks.

While the reviewers acknowledge the paper's novelty and appreciate its motivation, they disagree on the writing quality and find that the methods and results sections need significant improvement. The reviewers expressed serious concerns about the unconvincing experiments, which lack empirical evidence to support the proposed approach's superiority. Additionally, the baselines chosen for comparison don't include state-of-the-art methods, and the reviewers also found the ablation studies to be lacking.

Even if the authors provided feedback during the rebuttal stage, the modifications to the paper were minor.

Given the limited supporting evidence for the proposed method's advantages, I recommend rejecting the paper.

**Justification For Why Not Higher Score:**

The paper only presents limited empirical evidence supporting the method's advantages.

**Justification For Why Not Lower Score:**

N/A.

---

### Decision · Program_Chairs · 2024-01-16

Reject